# Genetic Biomarkers of Sorafenib Response in Patients with Hepatocellular Carcinoma

**DOI:** 10.3390/ijms25042197

**Published:** 2024-02-12

**Authors:** Lydia Giannitrapani, Francesca Di Gaudio, Melchiorre Cervello, Francesca Scionti, Domenico Ciliberto, Nicoletta Staropoli, Giuseppe Agapito, Mario Cannataro, Pierfrancesco Tassone, Pierosandro Tagliaferri, Aurelio Seidita, Maurizio Soresi, Marco Affronti, Gaetano Bertino, Maurizio Russello, Rosaria Ciriminna, Claudia Lino, Francesca Spinnato, Francesco Verderame, Giuseppa Augello, Mariamena Arbitrio

**Affiliations:** 1Institute for Biomedical Research and Innovation, National Research Council (CNR), 90146 Palermo, Italy; lydiagiannitp@gmail.com (L.G.); melchiorre.cervello@irib.cnr.it (M.C.); 2Department of Health Promotion, Mother and Child Care, Internal Medicine and Medical Specialties, University of Palermo, 90127 Palermo, Italy; francesca.digaudio@community.unipa.it (F.D.G.); aurelio.seidita@unipa.it (A.S.); maurizio.soresi@unipa.it (M.S.); marco.affronti@policlinico.pa.it (M.A.); 3Department of Experimental and Clinical Medicine, Magna Graecia University, 88100 Catanzaro, Italy; scionti@unicz.it (F.S.); nicolettastaropoli@gmail.com (N.S.); tassone@unicz.it (P.T.); tagliaferri@unicz.it (P.T.); 4Medical and Translational Oncology Unit, A.O.U. R. Dulbecco, 88100 Catanzaro, Italy; cilibertodomenico@hotmail.com; 5Department of Legal, Economic and Social Sciences, Magna Graecia University, 88100 Catanzaro, Italy; agapito@unicz.it; 6Department of Medical and Surgical Sciences, University Magna Graecia of Catanzaro, 88100 Catanzaro, Italy; cannataro@unicz.it; 7College of Science and Technology, Temple University, Philadelphia, PA 19122, USA; 8Villa Sofia-Cervello Hospital, C.O.U. Medical Oncology, 90146 Palermo, Italy; francescaspinnato0@gmail.com (F.S.); f.verderame@villasofia.it (F.V.); 9Hepatology Unit, A.O.U. Policlinico-San Marco, Department of Clinical and Experimental Medicine, University of Catania, 95123 Catania, Italy; gaetanobertinounict@gmail.com; 10Liver Unit of ARNAS Garibaldi-Nesima, 95100 Catania, Italy; mrussello@tiscali.it; 11Institute of Nanostructured Materials, National Research Council (CNR), 90146 Palermo, Italy; rosaria.ciriminna@cnr.it (R.C.); claudialino@gmail.com (C.L.); 12Institute for Biomedical Research and Innovation, National Research Council (CNR), 88100 Catanzaro, Italy

**Keywords:** genetic polymorphism, pharmacogenomics, genotyping, anticancer drugs

## Abstract

The identification of biomarkers for predicting inter-individual sorafenib response variability could allow hepatocellular carcinoma (HCC) patient stratification. SNPs in angiogenesis- and drug absorption, distribution, metabolism, and excretion (ADME)-related genes were evaluated to identify new potential predictive biomarkers of sorafenib response in HCC patients. Five known SNPs in angiogenesis-related genes, including *VEGF-A*, *VEGF-C*, *HIF-1a*, *ANGPT2*, and *NOS3*, were investigated in 34 HCC patients (9 sorafenib responders and 25 non-responders). A subgroup of 23 patients was genotyped for SNPs in ADME genes. A machine learning classifier method was used to discover classification rules for our dataset. We found that only the *VEGF-A* (rs2010963) C allele and CC genotype were significantly associated with sorafenib response. ADME-related gene analysis identified 10 polymorphic variants in *ADH1A* (rs6811453), *ADH6* (rs10008281), *SULT1A2/CCDC101* (rs11401), *CYP26A1* (rs7905939), *DPYD* (rs2297595 and rs1801265), *FMO2* (rs2020863), and *SLC22A14* (rs149738, rs171248, and rs183574) significantly associated with sorafenib response. We have identified a genetic signature of predictive response that could permit non-responder/responder patient stratification. Angiogenesis- and ADME-related genes correlation was confirmed by cumulative genetic risk score and network and pathway enrichment analysis. Our findings provide a proof of concept that needs further validation in follow-up studies for HCC patient stratification for sorafenib prescription.

## 1. Introduction

Hepatocellular carcinoma (HCC) accounts for about 90% of liver cancers. Most patients with HCC are diagnosed at an advanced tumor stage when treatment options are very limited. Until 2018, when lenvatinib was approved, sorafenib was considered the gold standard in the first-line setting for the treatment of advanced HCC [1]. However, choosing between these two agents remains challenging due to their low impact on survival and their similar and well-tolerated safety profiles [2]. Recently, the atezolizumab-bevacizumab combination has emerged as the first-line systemic treatment, but sorafenib is still a relevant choice for refractory patients and those ineligible for immunotherapy [3]. Sorafenib exerts its action through the inhibition of tumor cell proliferation and angiogenesis via the targeting of several oncogenic signaling pathways involving serine/threonine and tyrosine kinases (RAF1, BRAF, VEGFR 1, 2, 3, PDGFR, KIT, FLT3, FGFR1, and RET) [4,5]. However, sorafenib resistance remains a major challenge in improving the effectiveness of HCC treatment. The underlying mechanisms for inter-individual variability in response to therapy have not been fully elucidated, and no validated markers have been found that are capable of predicting clinical outcomes or sorafenib tolerability [6,7]. Thus, the identification of suitable biomarkers for patient stratification for sorafenib response in HCC may potentially help physicians in guiding the selection of tailored treatments.

HCC is a hypervascular tumor in which angiogenesis plays an important role in tumor growth and progression. Among others, VEGF/VEGFR, angiopoietin (ANGPT), endothelial nitric oxide synthase (eNOS or NOS3), and hypoxia-inducible factor-1α (HIF-1α) signaling play an important role in regulating tumor angiogenesis [8]. Single nucleotide polymorphisms (SNPs) in angiogenesis-related genes have been reported to influence outcomes in HCC patients treated with sorafenib [9,10,11].

In the present study, we selected 5 SNPs in these angiogenesis-related genes for the genotyping of 34 HCC patients, of which 9 showed response (responders) to therapy and 25 no response (non-responders). Additionally, in a subgroup of HCC patients, we evaluated 1931 SNPs and 5 copy number variations in 231 genes involved in drug absorption, distribution, metabolism, and excretion (ADME) using the DMET Plus microarray assay for the identification of new potential predictive biomarkers of response and outcome [12,13]. Through a model learning (ML) process, we proceeded to apply rules to classify all patients in terms of the detected SNPs and genotypes and according to sorafenib response in order to identify a predictive genetic signature that could allow the stratification of non-responder/responder patients to sorafenib for tailored prescriptions. The performance of the ML approach was tested through the Receiver Operating Characteristic (ROC) curves which supported the overall discriminatory power of identified SNPs as predictive and prognostic factors. Furthermore, the correlation between angiogenesis- and ADME-related genes was confirmed by a cumulative genetic risk score (GRS) and by network and pathway enrichment analysis, which demonstrated the association of 8/12 identified genes placed in topological key points of the interaction networks involved in several key common biological pathways correlated to HCC and sorafenib. Our findings should be considered as a “proof of concept” to be further validated in follow-up studies for the stratification of HCC patients towards the improvement of therapeutic choices.

## 2. Results

### 2.1. Patient Demographic and Clinical Characteristics

Appendix A shows some demographic and clinical characteristics of the patients studied. The median age was 73 ± 6 years (range 57–89 years). Men were more prevalent with respect to women (24 males and 10 females). Hepatitis virus C (HCV) infection was present in 19 cases (56%), HBV infection in 6 (18%), and alcoholic- and cryptogenetic/metabolic-related HCC was present in 3 (9%) and 6 (18%) cases, respectively. According to the Barcelona Clinic Liver Cancer (BCLC) classification, 3 patients were BCLC stage A (9%), 21 patients were BCLC stage B (62%), and 10 patients were BCLC stage C (29%). As for the Child–Pugh score, 26 patients (76%) were class A, 6 (18%) were class B, and 2 (6%) were class C. Alpha-fetoprotein (AFP) was ≥400 ng/dL in 4 cases (12%) and <400 ng/dL in 30 cases (88%). Twenty-two patients (65%) received locoregional treatments. Six patients (18%) had portal invasion. Finally, 19 (56%) patients maintained their starting dose of sorafenib, whereas 15 (44%) underwent dose reduction.

### 2.2. Allele and Genotype Distributions

For the study of angiogenesis-related genes, 34 patients (9 responders and 25 non-responders) were genotyped using the TaqMan allelic discrimination method. The distribution of rs2010963, rs4604006, rs12434438, rs55633437, and rs2070744 alleles and genotypes in responder (R) and non-responder (NR) groups is shown in Table 1. Only the C allele (*p* = 0.004) (Table 1), and CC genotype (*p* = 0.046) of VEGF-A (rs2010963) were significantly associated with sorafenib response.

The genotyping by DMET^TM^ SNP panel allowed the identification, among 1,936 markers, of 10 SNPs in seven genes as significantly associated with sorafenib response: *ADH1A* (alcohol dehydrogenase 1A), *ADH6* (alcohol dehydrogenase 6), *SULT1A2/CCDC101* (Sulfotransferase Family 1A Member 2)/CCDC101 (Coiled-Coil Domain-Containing Protein 101), *CYP26A1* (Cytochrome P450 Family 26 Subfamily A Member 1), *DPYD* (Dihydropyrimidine dehydrogenase), *FMO2* (Flavin Containing Dimethylaniline Monoxygenase 2), and *SLC22A14* (Solute Carrier Family 22 Member 14) (Table 2).

The heterozygous genotypes of *SLC22A14* rs171248, rs149738, and rs183574 (*p* = 0.007) and the homozygous genotypes AA in *SULT1A2/CCDC101* rs11401 (*p* = 0.019), *DPYD* rs2297595 (*p* = 0.011), *FMO2* rs2020863 (*p* = 0.004), TT in *DPYD*9* rs1801265 (*p* = 0.0257), as well as the GG in *ADH6* rs10008281 (*p* = 0.019) and *CYP26A1* rs7905939 (*p* = 0.027), showed a significant association to a lack of response to sorafenib. Instead, a significant correlation to sorafenib efficacy was found for the CC genotype in *ADH1A* rs6811453 (*p* = 0.005) and the heterozygous genotypes AG in *SULT1A2/CCDC101* rs11401 (*p* = 0.019), *DPYD* rs2297595 (*p* = 0.011), *FMO2* rs2020863 (*p* = 0.004), and CT in *DPYD*9* rs1801265 (*p* = 0.026). The rs11401 is located on the 16p11.2 region which contains a splice site encompassing the *SULT1A2* gene (500B Downstream Variant) and the *CCDC101* gene. In addition, the homozygous genotypes of *SLC22A14* rs171248 (TT), rs149738 (AA), and rs183574 (AA) were found to be significantly associated with responder patients (*p* = 0.001).

### 2.3. SNPs and Classification Rules Related to Sorafenib Response

Figure 1 shows the classification tree computed from the RandomTree’s classifier using the sorafenib dataset. Transforming the classification tree into classification rules (1–13) obtained by analyzing the input genotype dataset, as shown in Table 3, makes it more straightforward to analyze and understand the meaning of the multiple relations between the SNPs and genotypes responsible for a particular phenotype of sorafenib response.

We identified ten classification rules by which to discriminate patients belonging to the non-responder setting, and three rules for the responder ones, with an accuracy of 86.9565%: a subject could satisfy a rule only if a correspondence existed between their own genotype and detected SNPs, against every couple of alleles within of a rule. For instance, to verify whether a subject matched, i.e., rule 6 in Table 3, it was necessary that the SNPs (rs171248, rs6811453, rs2010963, rs12434438) assessed in the subject presented as detected genotypes (TT, CT, CC, GG), respectively. Thus, only the subjects that matched all the genotypes within a rule could be classified as a “non-responder” according to the matching rule. The AUC of the ROC curve further validates the power of RT to distingue between patients belonging to one of the two classes. The AUC computed from the ROC curve displayed in Figure 2 is equal to 0.8259, a value that confirms the capability of RT methods to avoid bias in separating responder from non-responder patients. 

Afterward, we examined the cumulative effects of SNPs obtained from the classification tree, developing a GRS by summing the number of response alleles [14,15,16]. The response-increasing alleles were attributed based on their greater frequency in response subjects according to the literature data for angiogenesis-related genes [8,10,16] and data obtained in the present study for ADME-related genes. The rs7905939 SNP was excluded from the analysis since a clear response allele was not identified. For each SNP, a score of 0 was defined for homozygous non-response alleles, 1 for heterozygous response and non-response alleles, and 2 for two homozygous response alleles. A higher mean GRS score was significantly associated with responders compared to non-responders when the sum of the 5 scores for the rs2010963, rs4604006, rs12434438, rs183574, and rs6811453 variants was considered for each patient (*p* = 0.008) (Appendix A). The mean of the gene count score was 6.00 ± 0.81 in the responder group, and 4.37 ± 1.36 in the non-responder group.

To explore whether the expression of angiogenesis- and ADME-related genes identified in the decision tree (i.e., *SLC22A4*, *ADH1A*, *VEGF-A*, *VEGF-C*, *HIF-1α*, and *CYP26A1*) might have a role in HCC disease outcome in terms of response to sorafenib, we carried out bioinformatic analysis of these genes using the public dataset GSE109211, downloaded from GEO, in which data from a subset of HCC patients (n = 67) treated with sorafenib are reported. As shown in Figure 3, VEGF-A, HIF-1α, and ADH1A expression were significantly lower in HCC tissues from sorafenib-responsive patients (n = 20), whereas SLC22A14 expression was significantly higher. No significant correlation was found between the expression of VEGF-C and CY26A1 genes and sorafenib response.

### 2.4. Pathway Enrichment Results

The network analysis highlighted a total of 14 genes with a relevant node degree score. Figure 4 shows the “seedGeneNetwork” with the key interactions among the 14 seed genes and other genes involved in several canonical pathways.

The seed genes enriching multiple pathways highlighted the relationships between seed genes and the affected biological functions involved in HCC and sorafenib. Among all genes, *ADH1A* and *CYP26A1*, along with *VEGF-A* and *VEGF-C*, showed a common involvement in the signal transduction pathways, which are reported as dysregulated in HCC, leading to uncontrolled cell division and metastasis [17].

Table 3 reports the top 21 degree-ranked seed genes among which, with a high degree value, are included 8/12 genes identified in the study (in bold). Analyzing Table 4, it is worth noting that a seed gene’s degree is higher than the added genes and refers to the centrality of a node in the network. The higher centrality of the degree reveals the crucial roles of genes in the network. Figure 5 shows the top ten degree-ranked seed genes computed using CytoHubba.

## 3. Discussion

HCC patients treated with sorafenib show a highly variable response, and patients experience resistance and adverse events in approximately 30% of cases. The molecular mechanisms underlying inter-individual variability in sorafenib response have yet to be fully elucidated, and a deeper knowledge of the underlying mechanisms and associated gene variants would allow us to tailor treatment prescriptions for better outcomes. 

Thus, with this aim, we studied 5 known candidate SNPs in genes controlling tumor angiogenesis, *VEGF-A* (rs2010963), *VEGF-C* (rs4604006), *HIF-1α* (rs12434438), *ANGPT2* (rs55633437), and *NOS3* (rs2070744), and genotyped a subgroup of 23 HCC patients using the DMET plus platform to identify potential prognostic biomarkers correlated to HCC patient responses to sorafenib treatment. Moreover, in a subgroup of patients (n = 11), the serum/plasma concentration of sorafenib was determined. The average steady-state sorafenib concentration was 4.13 mg/L in responders, and 4.41 mg/L in non-responders, independent of sorafenib dosages. No significant correlation was detected between sorafenib concentration and clinical outcome (data not shown), as already reported in the literature [18].

In our study, the analysis showed that the allele and genotype frequencies of SNPs in angiogenesis-related genes were significantly correlated with the response to sorafenib only for the rs2010963 C allele (*p* = 0.004) and the CC genotype (*p* = 0.046) of the *VEGF-A* gene, in accordance with the results of the retrospective multicenter study ALICE-2, in which the rs2010963 C allele and CC/CG genotype were significantly associated with a higher median overall survival of HCC patients receiving sorafenib [10]. It is likely that *VEGF-A*-related genetic variants could influence the level of circulating VEGF [19,20], therefore affecting sorafenib response. Also, in the SHARP trial [21], it was reported that a low VEGF-A plasma baseline level, as a prognostic independent factor, can predict outcomes in patients with advanced HCC, both in the entire patient population and in the placebo cohort [21]. Consistent with these results, in our GEO analysis, patients expressing lower levels of VEGF-A mRNA showed a better response to sorafenib therapy (GSE109211 dataset). Our findings confirmed the prominent role of the rs2010963 gene variant and VEGF-A expression as significant predictive factors for sorafenib response in HCC patients [10,22].

DMET genotyping showed a statistically significant association of the sorafenib “non-responder” phenotype with the heterozygous genotypes of *SLC22A14* rs171248, rs149738, and rs183574 and the homozygous genotypes AA in *SULT1A2/CCDC101* rs11401, *DPYD* rs2297595, *FMO2* rs2020863, the TT in *DPYD*9* rs1801265, the GG in *ADH6* rs10008281, and *CYP26A1* rs7905939. Instead, the sorafenib “responder” phenotype was associated with the genotypes CC in *ADH1A* rs6811453, the AG in *SULT1A2/CCDC101* rs11401, *DPYD* rs2297595, *FMO2* rs2020863, and CT in *DPYD*9* rs1801265, as well as with the homozygous genotypes of *SLC22A14* rs171248 (TT), rs149738 (AA), and rs183574 (AA). These results demonstrate that the ADME genotype is correlated with different responses to sorafenib, underlying the role of the reference allele or variant in the effect on treatment response. 

Moreover, to verify whether a correlation of SNPs in angiogenesis- and ADME-related genes might help to discriminate responder/non-responder patients, we applied a classifier to mine classification rules able to figure out the principal signatures for discriminating among patients belonging to the responder/non-responder to sorafenib phenotypes. The novelty of our study lies in the identification of 10 rules in different genotype associations for the identification of the non-responder phenotype, and 3 rules for the responder type. These rules may represent a genetic signature that could allow the stratification of patients who are fit for sorafenib treatment. We found that the genetic signature including 3 ADME-SNPs, *SLC22A14* (rs171248), *ADH1A* (rs6811453), and *CYP26A1* (rs7905939) and 3 known SNPs in angiogenesis-related genes, *VEGF-A* (rs2010963), *VEGF-C* (rs4604006), and *HIF-1A* (rs12434438), was correlated to sorafenib response in our dataset, allowing the discrimination between responders/non-responders according to mRECIST criteria. The signature of the response, identified by a decision tree, was also validated by the GRS analysis.

However, while the role of angiogenesis-related genes is well known in HCC patients treated with sorafenib, little information is reported in the literature regarding the role of selected ADME genes in this context. *SLC22A14*, also known as organic cation transporter-like 2 (*OCTL2*), is a gene encoding a member of the organic-cation transporter family and anions (OATs), whose expression is high in the liver. As for the other SLCs, it is involved in regulating the homeostasis of metabolites, the uptake of a wide range of molecules, and the disposition of drugs, as well as in promoting cell proliferation, migration, and invasion in HCC [23]. The function of SLC members in sorafenib resistance is not clear, and only recently have studies begun to investigate their role in chemoresistance [24,25,26,27], highlighting the role of aberrant variants or SNPs in organic cation transporters during liver carcinogenesis, with effects on the ability of HCC to respond to sorafenib.

The *ADH1A* gene catalyzes the oxidation of alcohols to aldehydes and belongs to the superfamily of dehydrogenase enzymes. Several reports provide a correlation of ADH1A and other ADHs’ expression with increased risk of liver cancer, with an impact on the prognosis for HCC patients [28].

*CYP26A1*, a member of the cytochrome P450 enzyme superfamily, is mainly involved in retinoic acid metabolism and the synthesis of cholesterol, steroids, and other lipids, and it contributes to the development and progression of multiple cancers [29,30,31]. Previous studies in HCC have shown that CYP26A1 mRNA is downregulated in tumor tissue compared to paired-matched non-tumor tissues [32], but the role of *CYP26A1* in HCC is not entirely clear despite being reported as hypovitaminosis. A decrease in retinoic acid, as a potential result of a CYP26A1 depletion, could be correlated to a higher risk of carcinogenesis [33].

To support the correlation between ADME- and angiogenesis-related genes, the network and PEA analysis highlighted the association of 8/12 identified genes in topological key points with a relevant node degree score in important pathways underlying biological mechanisms implicated in HCC and sorafenib. In fact, *VEGF-A*, together with *ADH1A*, *CYP26A1*, and *VEGF-C*, showed a common interaction in “signal transduction pathways” which are known to be dysregulated in HCC, with consequent uncontrolled cell division and metastasis [27], alteration of intracellular regulators or extracellular signals with abnormal epigenetic modification, and gene expression in the tumor microenvironment. Moreover, all seed genes identified were involved in multiple pathways, showing their involvement is significantly affected by biological functions.

Given that the most recent AASLD guidelines on HCC systemic therapy have approved the use of drugs like bevacizumab + atezolizumab and lenvatinib as first-line therapies, in addition to sorafenib, the opportunity to verify a score like the GRS observed in our responder patients or the genetic signature identified in our retrospective study might help clinicians to select patients with higher chances to benefit from sorafenib treatment. In the presence of a favorable GRS, physicians could treat advanced HCC patients with sorafenib as soon as possible. Conversely, patients with an unfavorable GRS might not be excellent candidates for sorafenib. 

## 4. Materials and Methods

### 4.1. Study Participants 

Thirty-four patients with advanced HCC who had undergone sorafenib treatment were enrolled in our study and distributed into two groups (responders vs. non-responders) according to modified Response Evaluation Criteria in Solid Tumors (mRECIST) [34]. Among them, patients who showed partial response or stable disease (SD) for more than 6 months were classified as responders, while patients with a progressive disease were considered non-responders. Only two patients showed a complete response. We classified patients from 5 medical centers: the Department of Health Promotion, Mother and Child Care, Internal Medicine and Medical Specialties, University Hospital of Palermo, Italy; the Oncology Unit, AOU Mater Domini, Catanzaro, Italy; the Department of Clinical and Experimental Medicine Policlinico “G. Rodolico” University of Catania, Italy; the Liver Unit of ARNAS Garibaldi-Nesima, Catania, Italy; the C.O.U. Medical Oncology, Villa Sofia-Cervello Hospital, Palermo, Italy. All patients were enrolled according to inclusion criteria provided by the clinical study protocol. More in detail, since the regulatory approval of the use of sorafenib as a drug for the treatment of HCC is not amenable to locoregional therapy, all the patients who met the inclusion criteria according to international guidelines (AASLD and EASL) [35,36] for medical therapy, and had therefore started therapy with sorafenib, were included in the study after signing the informed consent.

The study was approved by the Ethics Committees of the University Hospital of Palermo, and the Ethics Committees of Section Area Centro (Region of Calabria), as spontaneous study No. 3/2017 and Prot. n. 387, respectively. All patients gave their approval and signed informed consent according to the recommendations of the Declaration of Helsinki for biomedical research involving human subjects [37].

### 4.2. Sample Collection and Genotyping

For each subject, a 5 mL peripheral blood sample was collected in EDTA anticoagulant tubes. The sample was centrifuged, and then the plasma was replaced with an equal amount of physiological solution (0.9% sodium chloride) and stored at −80 °C. Genomic DNA was extracted from whole blood using QIAamp DNA Blood Mini Kit (Qiagen, Valencia, CA, USA), in accordance with the manufacturer’s instructions. For the evaluation of selected SNPs in angiogenesis-related genes, DNA samples of 34 HCC patients were genotyped using the TaqMan allelic discrimination method (StepOne Plus™ Real-Time PCR System, Applied Biosystems, Monza, Italy) using commercial (*VEGF-A* rs2010963, *VEGF-C* rs4604006, *HIF-1A* rs12434438, *ANGP2* rs55633437, *NOS3* rs2070744) genotyping assays (ThermoFisher Scientific, Waltham, MA, USA). DMET Plus assay (ThermoFisher Scientific, Waltham, MA, USA) was performed as previously described [38,39,40].

### 4.3. Genetic Risk Score

To build a GRS for sorafenib, we estimated the genetic profiles of patients depending on frequencies of genotypes and alleles of examined gene polymorphisms. For each SNP, a score of 0 was defined for homozygous non-responder alleles, 1 for heterozygous responder and non-responder alleles, and 2 for two homozygous responder alleles [14,15,16]. The response-increasing alleles were attributed based on their greater frequency in response subjects according to the literature data for angiogenesis-related genes [9,10,11,41] and by data obtained in the present study for ADME-related genes. For each subject, a combined GRS was calculated as the sum of these response-increasing alleles; the score was 0–10 for 5 variants.

### 4.4. Statistical Analysis

Normally distributed continuous variables were expressed as mean ± SD, and the differences between the two groups were assessed with the Mann–Whitney U test. We used the G*Power software tool to evaluate the statistical Power of the analysis. G*Power is a tool to compute statistical power analyses for many statistical tests. In order to determine the number of participants needed for a study, we used G*Power and set the input parameters accordingly. We used χ^2^ tests for Goodness-of-fit tests, Contingency tables, and analysis a priori to compute the required sample size. We configured the input parameters of G*Power as follows: Effect Size w = 0.8; α Error Probability = 0.05; Power (1-β error probability) = 0.8; Degree of Freedom Df = 5. Based on these parameters, we found that a total sample size of 21 was required to obtain a Statistical Power of 0.82. We used 23 samples, which is consistent with the value calculated by G*Power. Therefore, we can confidently conclude that the Statistical Power of our study is at least 0.82.

DMET analysis of frequencies was calculated using Fisher’s exact test and Bonferroni’s correction for multiple comparisons using the DMET-Analyzer tool [12]. In all analyses, a *p*-value < 0.05 was considered statistically significant. ADME genotyping calls from the intensity array were performed with DMET Console software (version 1.1, ThermoFisher Scientific, Waltham, MA, USA). All alleles were tested for Hardy–Weinberg equilibrium. Genotypes with a call rate ≤ 96%, a high rate of “possible rare allele”/“no call”, and a concordance of 100% were excluded from further analysis.

### 4.5. Classification Rules

To discover classification rules for our dataset, starting from angiogenesis-related SNPs and identified SNPs in ADME genes (sorafenib dataset), we applied a classifier for the identification of rules that could facilitate the stratification of responder/non-responder patients in terms of sorafenib response. Classifiers belong to the non-parametric supervised learning algorithm category, where the machines learn patterns buried in the data using previously labeled training data which is mandatory for supervised learning. Supervised learning aims to produce a model that predicts class variable values by learning simple patterns inferred from the data features. A RandomTree (RT) is a single decision-making entity that works by splitting the data into subsets based on the value of input features. This process is repeated recursively, resulting in a tree with decision and leaf nodes. The main features of a RT include its simplicity and interpretability which allow an easy visualization and understanding of how the decisions are made. We used the Random Tree classifier available in the Weka ver. 3.8.6 software framework to perform the analysis, applying the 5-fold cross-validation mode as the test mode, reaching an accuracy of 86.9565. Moreover, to evaluate the performance of an RT classifier, we used the ROC (Receiver Operating Characteristic) curves, which measure the trade-off between sensitivity (or True Positive Rate) and specificity (1—False Positive Rate) at various threshold settings. The ROC curve is a powerful tool for assessing the performance of binary classification models, providing insights into how well the model can distinguish between the two classes under various threshold settings. The RT’s output is a classification tree easily understandable even by non-domain experts which can be quickly translated into classification rules in the “IF (cond1 & cond2 & … & cond n) THEN class” format. 

The classification rules can be used to categorize “in general” any genotypes obtained from subjects treated with sorafenib and genotyped using DMET-plus microarray.

### 4.6. Network and Pathway Enrichment Analysis

Network analysis was used to determine the influential role of the identified seed genes within the population affected by HCC and treated with sorafenib.

To consolidate the seed genes, we conducted a gene network consolidation approach like the one proposed by Agapito et al. [42]. The consolidation approach consists of (i) mapping seed genes on the human protein–protein interaction (PPI) network from the Integrated Interactions Database (IID) [43]; (ii) for each mapped seed gene, the expansion process regards the computing of the community with a radius equal to one, allowing the collection of all genes that share similar functions or involvement in similar biological processes with seed genes [44]. We used Cytoscape [45] and the Cytoscape plugin CytoHubba [46] to assess the mapped seed genes’ roles in the “seedGeneNetwork,” computing the degree of the seed genes. The node degree is the sum of all edges connected to it. A node with a degree equal to d indicates that the neighbor radius nodes linking it is d. To further investigate the seed genes list, we computed pathway enrichment analysis to gain insight into the affected unknown underlying biological mechanisms associated with sorafenib response in the 34 HCC patients.

Reactome enrichment analyses were performed using BiP and pathDIP [47,48].

## 5. Conclusions

The novelty of our study is the identification of genetic signatures through classification rules for SNPs in angiogenesis and ADME genes associated with sorafenib responses in HCC patients, through which it may be possible to personalize prescription. The application of GRS could allow a better stratification of patients. In addition, the network analysis conducted in this study supports the association of 8/12 analyzed genes in topological key points involved in several common biological pathways correlated to HCC and sorafenib. However, our study has some limitations. The sample size was relatively small, and further investigations in a larger sample size may be needed. The opportunity to test the classification rules on predictive biomarkers of response to sorafenib in an independent and larger validation set would give more robustness to our findings. Therefore, our findings had an exploratory aim and are intended as a “proof-of-concept” research to be further validated in a larger dataset to allow sorafenib-tailored prescriptions through predictive biomarkers of response and outcome in HCC patients. The involvement of seed genes in multiple biological pathways related to sorafenib and HCC, as well as the common interactions of *ADH1A*, *CYP26A1*, *VEGF-A*, and *VEGF-C* in signal transduction pathways, should allow future studies on the simultaneous targeting of different signaling pathways or common downstream proteins involved in HCC control and sorafenib response with the aim of personalizing treatment for this still uncurable disease.

## Figures and Tables

**Figure 1 ijms-25-02197-f001:**
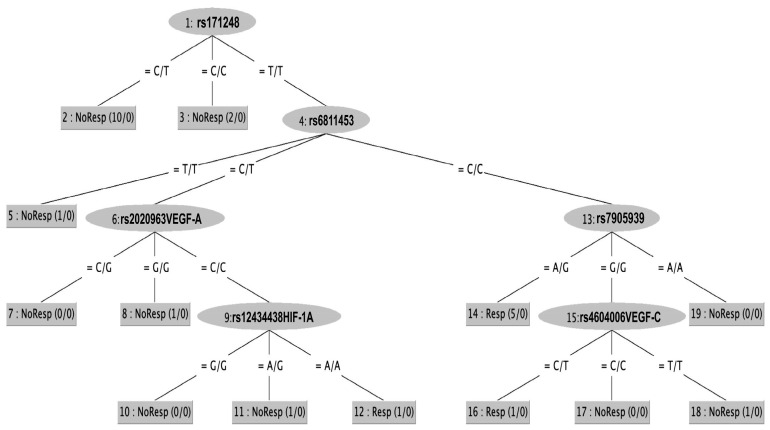
The Decision Tree is computed from the RandomTree classifier, available in Weka, using the sorafenib dataset. The RandomTree’s parameters were set up as follows: weka.classifiers.trees. RandomTree -K 0 m 1.0 -V 0.001 -S 1, the selected Test models 10-fold cross-validation, reaching an accuracy of 86.9565%.

**Figure 2 ijms-25-02197-f002:**
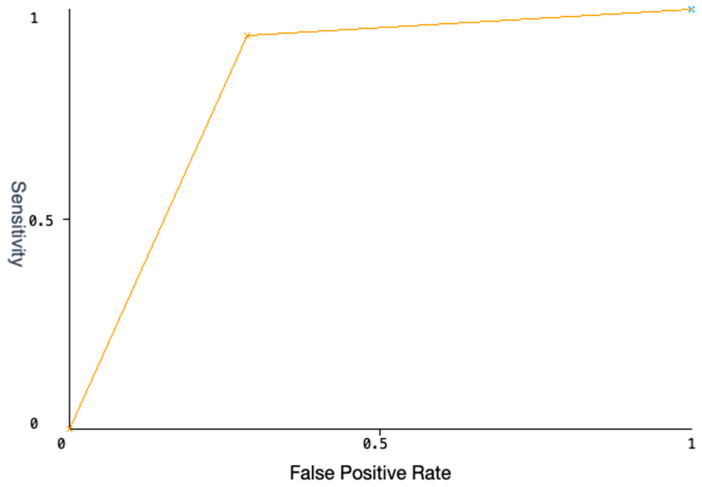
The ROC curve displays the trade-off between the True Positive Rate (TPR) and False Positive Rate (FPR) across different thresholds, illustrating a binary classifier’s performance. The curve’s proximity to the top-left corner indicates better model accuracy, with the Area Under the Curve (AUC) metric summarizing overall effectiveness (0.8259). The *y*-axis represents the sensitivity while the *x*-axis represents the False Positive Rate.

**Figure 3 ijms-25-02197-f003:**
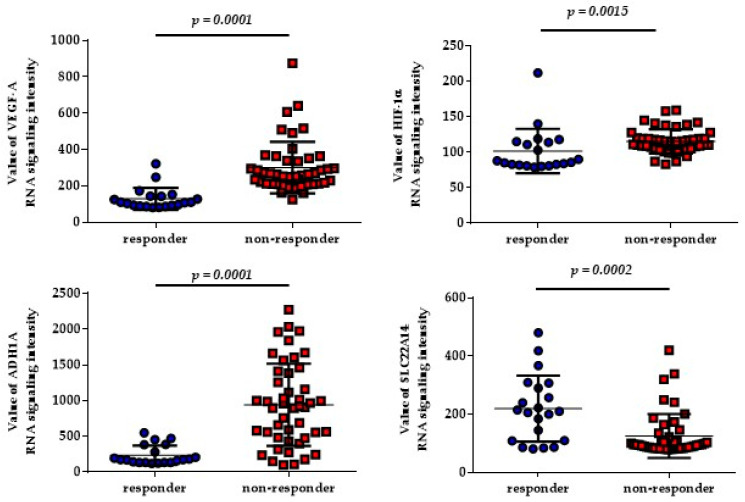
Expression of genes identified by decision tree analysis in HCC patients treated with sorafenib. Data are expressed as mean ± SD, and the differences between the two groups were assessed with the Mann–Whitney U test.

**Figure 4 ijms-25-02197-f004:**
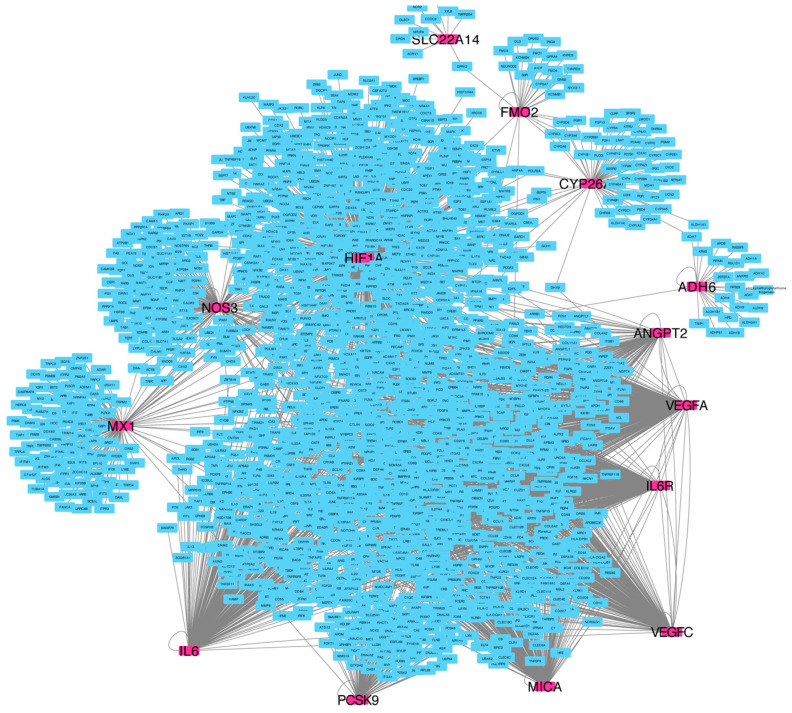
The consolidated network was obtained for each seed gene by computing the neighborhoods with a radius of one.

**Figure 5 ijms-25-02197-f005:**
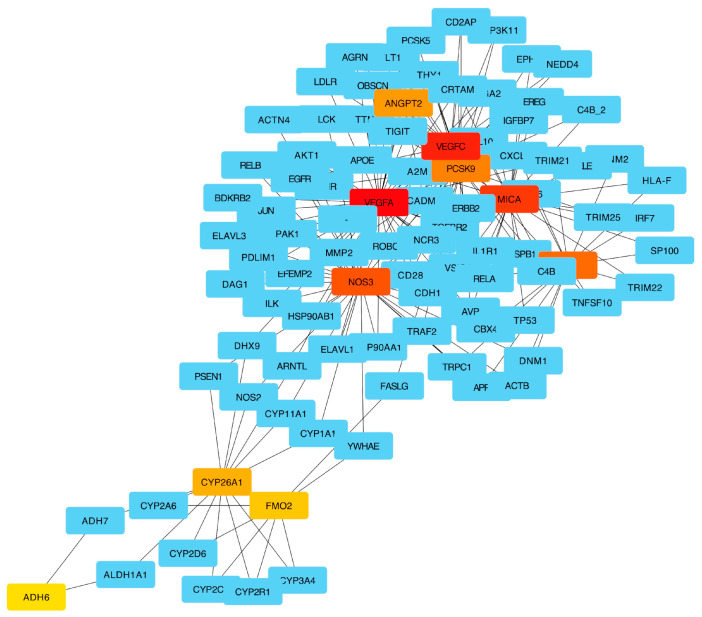
Top 10 genes by their degree of relevance in the seed network, computed using CytoHubba: high significance values are represented by red, orange, and yellow, while all bluish colors represent less significant values.

**Table 1 ijms-25-02197-t001:** Association between genetic variants and sorafenib response in HCC patients (n = 34).

SNP (Gene)	R n (%)	NR n (%)	*p*
**rs2010963 (*VEGF-A*)**			
**Allele**			
G	2 (11.1)	25 (50.0)	-
C	16 (88.9%)	25 (50.0)	**0.004**
**Genotype**			
GG	0 (0.0)	7 (28.0)	-
GC	2 (22.2)	11 (44.0)	0.056
CC	7 (77.8)	7 (28.0)	**0.046**
**rs4604006 (*VEGF-C*)**			
**Allele**			
C	13 (72.2)	34 (68.0)	-
T	5 (27.8)	16 (32.0)	0.739
**Genotype**			
CC	5 (57.0)	11 (44.0)	-
CT	3 (28.5)	12 (48.0)	0.474
TT	1 (14.5)	2 (8.0)	0.943
**rs12434438 (*HIF-1α*)**			
**Allele**			
A	14 (77.8)	30 (60.0)	-
G	4 (22.2)	20 (40.0)	0.175
**Genotype**			
AA	5 (55.6)	9 (36.0)	-
AG	4 (44.4)	12 (48.0)	0.522
GG	0 (0.0)	4 (16.0)	0.159
**rs55633437 (*ANGPT2*)**			
**Allele**			
G	16 (88.9)	38 (76.0)	-
T	2 (11.1)	12 (24.0)	0.246
**Genotype**			
GG	7 (77.8)	14 (56.0)	-
GT	2 (22.2)	10 (40.0)	0.301
TT	0 (0.0)	1 (4.0)	0.484
**rs2070744 (*NOS3*)**			
**Allele**			
C	6 (33.3)	19 (38.0)	-
T	12 (66.7)	31 (62.0)	0.724
**Genotype**			
CC	0 (0)	4 (16.0)	-
CT	6 (66.7)	11 (44.0)	0.159
TT	3 (33.3)	10 (40.0)	0.289

Bold indicates statistical association (*p* ≤ 0.05). R = responders; NR = non-responders.

**Table 2 ijms-25-02197-t002:** ADME SNPs correlated to sorafenib response in HCC patients (n = 23).

SNP (Gene)	R n (%)	NR n (%)	*p*
**rs6811453** **(*ADH1A*)**			
**Allele**			
T	1 (7.2)	18 (56.2)	-
C	13 (92.8)	14 (43.8)	**0.002**
**Genotype**			
TT	0 (0.0)	5 (24.0)	-
CT	1 (14.0)	8 (48.0)	-
CC	6 (86.0)	3 (28.0)	**0.005**
**rs10008281 (*ADH6*)**			
**Allele**			
G	6 (42.8)	28 (87.5)	-
T	8 (57.2)	4 (12.5)	**0.003**
**Genotype**			
TT	2 (28.6)	0 (0.0)	-
GT	4 (57.1)	4 (25.0)	-
GG	1 (14.3)	12 (75.0)	**0.019**
**rs11401 (*SULT1A2*)**			
**Allele**			
A	7 (50.0)	27 (84.3)	-
G	7 (50.0)	5 (15.7)	**0.026**
**Genotype**			
AA	1 (15.5)	12 (75.0)	**0.019**
AG	5 (71.0)	3 (18.7)	**0.019**
GG	1 (14.5)	1 (6.3)	-
**rs7905939 (*CYP26A1*)**			
**Allele**			
G	6 (42.8)	6 (18.7)	-
A	8 (57.2)	26 (81.3)	0.143
**Genotype**			
AA	0 (0.0)	1 (6.3)	-
AG	6 (85.7)	4 (25.0)	-
GG	1 (14.3)	11 (68.7)	**0.027**
**rs2297595 (*DPYD*)**			
**Allele**			
G	9 (64.2)	30 (93.7)	-
A	5 (35.8)	2 (6.3)	**0.020**
**Genotype**			
AA	2 (28.6)	14 (87.5)	**0.011**
AG	5 (71.4)	2 (12.5)	**0.011**
GG	0 (0.0)	0 (0.0%)	-
**rs1801265 (*DPYD*)**			
**Allele**			
C	9 (64.2%)	29 (90.6)	**0.044**
T	5 (35.8)	3 (9.4)	-
**Genotype**			
TT	2 (28.6)	13 (81.3)	**0.026**
CT	5 (71.4)	3 (18.7)	**0.026**
CC	0 (0.0)	0 (0.0)	-
**rs2020863 (*FMO2*)**			
**Allele**			
G	4 (28.5)	0 (0.0)	**0.006**
A	10 (71.5)	32 (100.0)	-
**Genotype**			
AA	3 (42.9%)	16 (100.0)	**0.004**
AG	4 (57.1)	0 (0.0)	**0.004**
GG	0 (0.0)	0 (0.0)	-
**rs171248 (*SLC22A14*)**			
**Allele**			
T	14 (100.0)	18 (56.2)	**0.004**
C	0 (0.0)	14 (43.8)	-
**Genotype**			
TT	7 (100.0)	4 (25.5)	**0.001**
CT	0 (0.0)	10 (62.5)	**0.007**
CC	0 (0.0)	2 (12.5)	-
**rs149738 (*SLC22A14*)**			
**Allele**			
T	14 (100.0)	18 (56.2)	**0.004**
C	0 (0.0)	14 (43.8)	-
**Genotype**			
TT	7 (100.0)	4 (25.5)	**0.001**
CT	0 (0.0)	10 (62.5)	**0.007**
CC	0 (0.0)	2 (12.5)	-
**rs183574 (*SLC22A14*)**			
**Allele**			
C	0 (0.0)	14 (43.7)	-
T	14 (100.0)	18 (56.3)	**0.004**
**Genotype**			
TT	7 (100.0)	4 (25.5)	**0.001**
TC	0 (0.0)	10 (62.5)	**0.007**
CC	0 (0.0)	2 (12.5)	-

**Table 3 ijms-25-02197-t003:** Classification rules related to sorafenib response: The IF-THEN form, obtained from the classification tree, is used to classify patients (n = 23) into responder (R) and non-responder (NR) to sorafenib treatment.

If rs171248 = CT then NR If rs171248 = CC then NR If rs171248 = TT & rs6811453 = TT then NR If ss171248 = TT & rs6811453 = CT & rs2010963 = CG then NR If rs171248 = TT & rs6811453 = CT & rs2010963 = GG then NR If rs171248 = TT & rs6811453 = CT & rs2010963 = CC & rs12434438 = GG then NR If rs171248 = TT & rs6811453 = CT & rs2010963 = CC & rs12434438 = AG then NR If rs171248 = TT & rs6811453 = CC & rs7905939 = GG & rs4604006 = CC then NR If rs171248 = TT & rs6811453 = CC & rs7905939 = GG & rs4604006 = TT then NR If rs171248 = TT & rs6811453 = CC & rs7905939 = AA then NR
If rs171248 = TT & rs6811453 = CT & rs2010963 = CC & rs12434438 = AA then R If rs171248 = TT & rs6811453 = CC & rs7905939 = AG then R If rs171248 = TT & rs6811453 = CC & rs7905939 = GG & rs4604006VEGF-C = CT then R

rs171248 is on the *SLC22A14* T>C gene, rs6811453 is on *ADH1A*_c.*2804 C>T, rs7905939 is on *CYP26A1*_c.*4046 G>A, rs2010963 is on *VEGF-A*, rs4604006 is on *VEGF-C*, and rs12434438 is on *HIF-1α*.

**Table 4 ijms-25-02197-t004:** Top 21 degree-ranked seed genes. The genes studied are in bold. The degree was computed using CytoHubba.

Rank	Name	Score
**1**	** *VEGFA* **	730,700.53
**2**	*MICA*	379,197.11
**3**	** *NOS3* **	339,569.49
**4**	** *VEGFC* **	313,274.47
**5**	*MX1*	249,727.84
**6**	*PCSK9*	194,571.69
**7**	** *CYP26A1* **	79,311.53
**8**	** *ANGPT2* **	112,225.26
**9**	** *FMO2* **	69,997.19
**10**	*DHX9*	65,954.87
**11**	*ARNTL*	65,954.87
**12**	** *ADH6* **	51,195.00
**13**	*RELA*	47,118.36
**14**	*CXCL10*	43,934.98
**15**	*FASLG*	32,946.27
**16**	*YWHAE*	29,395.65
**17**	*ALDH1A1*	26,473.00
**18**	*ADH7*	26,473.00
**19**	*GPR12*	23,300.00
**20**	*TGFBR2*	21,738.55
**21**	** *SLC22A14* **	21,060.00

## Data Availability

Data are contained within the article and Appendix A.

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
