# Peer review of "Genetic Biomarkers of Sorafenib Response in Patients with Hepatocellular Carcinoma"

_ijms, 2024, doi:10.3390/ijms25042197_

Round 1
Reviewer 1 Report
Comments and Suggestions for Authors
Authors have used a ML approach based on tree's classifier, it have named it "random tree", but is not it "random forest"?
the details of the method of ML along the name of the software should be described in the methods section. The accuracy, precision and CV results has to be reported such as ROC curve.
Reviewer 2 Report
Comments and Suggestions for Authors
If the biomarkers shown in this analysis by the authors are valid, this is an interesting report that may have an effect on personalized medicine in the treatment of sorafenib.
I have several concerns about this paper.
Major comments
1) As stated by the authors, the reliability of this analysis is questionable because the sample size is too small to show statistically significant differences. How good was the statistical power (power of detection) in this analysis?
2) The author selected five SNPs among angiogenesis-related genes. Why were these selected from among the many reported angiogenesis-related genes and SNPs?
3) Figure 2 shows the results of analysis using the public data GSE109211, but is there any comparison of the expression levels of angiogenesis- and ADME-related genes in the authors' patient groups? Wouldn't the presentation of raw data be more convincing in the original paper?
4) Are the 10 classification rules presented by the authors worth using in general or are they acceptable as rules in this paper?
Minor comments
1) Page 6, line 162 and line 166. correct HIF-1 to HIF-1a.
2) Page 11, line 284. What does A mean in the sentence "A, as a potential result from a CYP26A1 depletion, could be correlated to a higher risk of carcinogenesis.
Reviewer 3 Report
Comments and Suggestions for Authors
The authors present an interesting topic of hepatology. The novelty of their study is the identification of genetic signatures through classification rules for SNPs in angiogenesis and ADME genes associated with sorafenib responses in HCC patients. It is an important task of personalized medicine. However, the sample (34, and 23 specimens) is very small and may not reach the significance to be cited in hepatology platforms. I was wondering if the authors could add a graphical abstract including the supposed mechanisms. The figures and the details presented are of low quality and need significant improvement.
Comments on the Quality of English LanguageMinors.
Round 2
Reviewer 2 Report
Comments and Suggestions for Authors
The authors' post-edited paper appears to be a proper response to the reviewer's comments.